# CALPHAD accelerated design of advanced full-Zintl thermoelectric device

Li Yin ⬤ [1,2,6], Xiaofang Li [2,6], Xin Bao[1], Jinxuan Cheng[1], Chen Chen[3], Zongwei Zhang[4], Xingjun Liu[1,5], Feng Cao ⬤ [2], Jun Mao ⬤ [1,5] & Qian Zhang ⬤ [1,5] ✉

Since thermoelectric materials have different physical and chemical properties, the design of contact layers requires dedicated efforts, and the welding temperatures are distinctly different. Therefore, a general interface design and connection technology can greatly facilitate the development of thermoelectric devices. Herein, we proposed a screening strategy for the contact materials based on the calculation of phase diagram method, and $Mg_2Ni$ has been identified as a matched contact layer for n-type $Mg_3Sb_2$-based materials. And this screening strategy can be effectively applied to other thermoelectric materials. By adopting the low-temperature sintering silver nanoparticles technology, the Zintl phase thermoelectric device can be fabricated at low temperature but operate at medium temperature. The single-leg n-type $Mg_{3.15}Co_{0.05}SbBi_{0.99}Se_{0.01}$ device achieves an efficiency of ~13.3%, and a high efficiency of ~11% at the temperature difference of 430 K has been realized for the Zintl phase thermoelectric device comprised together with p-type $Yb_{0.9}Mg_{0.9}Zn_{1.198}Ag_{0.002}Sb_2$. Additionally, the thermal aging and thermal cycle experiments proved the long-term reliability of the $Mg_2Ni/Mg_{3.15}Co_{0.05}SbBi_{0.99}Se_{0.01}$ interface and the nano-silver sintering joints. Our work paves an effective avenue for the development of advanced devices for thermoelectric power generation.

Thermoelectric (TE) devices are capable of converting dispersed waste heat into electricity to achieve the full utilization of energy and support the global sustainable development[1–3]. A prospective TE device should possess high efficiency, low cost, and long-term stability. Among the state-of-the-art TE materials, Zintl phase materials are one of the most promising TE materials with excellent TE performance, together with abundant and environmentally friendly constituent elements[4,5].

To facilitate the fabrication of high-performance devices, the design of two critical layers (i.e., contact and bonding layers) is of great importance. The contact layers are usually introduced to connect TE materials and the metallization layers, and the bonding layers are needed to connect the metallization layer and electrodes. They are of great significance in reducing the additional electrical and thermal resistance as well as the potential thermal stress, thereby greatly improving the performance and reliability of TE devices. Presently, due to the lack of reliable design for these interfaces, the efficiency of the Zintl phase device is low and cannot meet the requirements of the application[6,7]. Furthermore, the activity of constituent elements and the high coefficient of thermal expansion (CTE) further complicate the fabrication, hence delaying the development of highly efficient and stable Zintl phase devices[8,9].

Traditionally, the selection of contact materials has been based on an empirical method, including the selection of high-melting-point

[1]School of Materials Science and Engineering, Harbin Institute of Technology, Shenzhen 518055, P.R. China. [2]School of Science, Harbin Institute of Technology, Shenzhen 518055, P.R. China. [3]School of Physical Sciences, Great Bay University, Dongguan 523000, P.R. China. [4]Ningbo Institute of Materials Technology and Engineering, Chinese Academy of Sciences, Ningbo 315201, P.R. China. [5]State Key Laboratory of Advanced Welding and Joining, Harbin Institute of Technology, Harbin 150001, P.R. China. [6]These authors contributed equally: Li Yin, Xiaofang Li. ✉e-mail: zhangqf@hit.edu.cn

metals to reduce reaction rates, matched CTE to improve mechanical reliability, and suitable work function to reduce contact resistance[10]. However, this trial-and-error method of screening is time- and cost-consuming and difficult to guide other materials systems. Phase diagram engineering has long been applied to develop various functional materials, and has recently been adopted to design the compositions and microstructures of TE materials[11–14]. The CALPHAD (CALculation of PHAse Diagram) method describes the thermodynamic properties of materials quantitatively using the Gibbs energy of each phase with corresponding parameters and simulates the kinetic process by solving diffusion equations numerically[15,16], which is critical for designing heterogeneous interfaces. In addition, to reduce the performance loss of electronic devices caused by the welding process, a variety of new connection technologies have been studied[17–20]. With the advantages of low-temperature connection and high-temperature service, low-temperature silver nanoparticles (Ag NPs) sintering has been applied to various functional material devices[19,21], and has now been adopted to fabricate TE devices[22–24].

In this work, the CALPHAD method was employed to screen the contact materials, and Ag NPs were used to realize the bonding connection. By using the CALPHAD method, the thermal stability between the TE material and other phases at a specific temperature can be evaluated, and it can guide an efficient screening of the possible interface materials. As an example, $Mg_2Ni$ has been identified as the contact layer for the n-type $Mg_{3.15}Co_{0.05}SbBi_{0.99}Se_{0.01}$. Combining the low-temperature Ag NPs sintering technology, a full Zintl phase device is fabricated. Advanced high conversion efficiency of ~11% in the single-stage Zintl phase device is achieved, and satisfactory long-term stability is realized. This study proves the feasibility of the CALPHAD method in screening the contact materials, and the reliability of

low-temperature Ag NPs sintering for TE devices, which can be applied to the design and fabrication of other TE devices.

## Results and discussion

A multiple-layer structure, including TE legs, barrier layers (screened by CALPHAD, Fig. 1a), solder layers (Ag NPs, Fig. 1c), electrode layers, and pre-circuited AlN ceramic plates were constructed for the device fabrication, as demonstrated in Fig. 1b. To design the contact layers with low contact resistance and good thermal stability, an efficient screening strategy was proposed by considering the diffusion doping effect together with thermodynamic stability. In view of previous in-depth studies on element doping effect[25–30], we can quickly identify potential congener doping or inert doping of single elements. Then, using CALPHAD to calculate the target phase diagram, candidate interface materials could be screened out according to the thermodynamic phase equilibrium between the TE material and other phases. Figure 1a shows the isothermal section at 773 K of the Mg-Ni-Sb ternary phase diagram that was obtained by the CALPHAD method based on the involved binary phase diagram and its thermodynamic data. Apparently, $Mg_2Ni$ is predicted as one of the promising candidates for a barrier layer for n-type $Mg_3Sb_2$ alloy, which consists of n-type doping elements and is thermodynamically stable to this alloy.

Herein, $Mg_{3.15}Co_{0.05}SbBi_{0.99}Se_{0.01}$ (n-Zintl) and $Yb_{0.9}Mg_{0.9}Zn_{1.198}Ag_{0.002}Sb_2$ (p-Zintl)[31] are chosen as the n-type and p-type leg, respectively. Detailed TE properties are presented in Figs. S1, S2. A high average dimensionless TE figure-of-merit ($zT$) value of ~1.34 between 300 K and 700 K was achieved by the regulation of Mg vacancies and Bi alloying in n-Zintl. Since the welding temperature is normally 100–300 K higher than the designed hot-side temperature, we annealed these two samples at 973 K for 20 min to simulate the

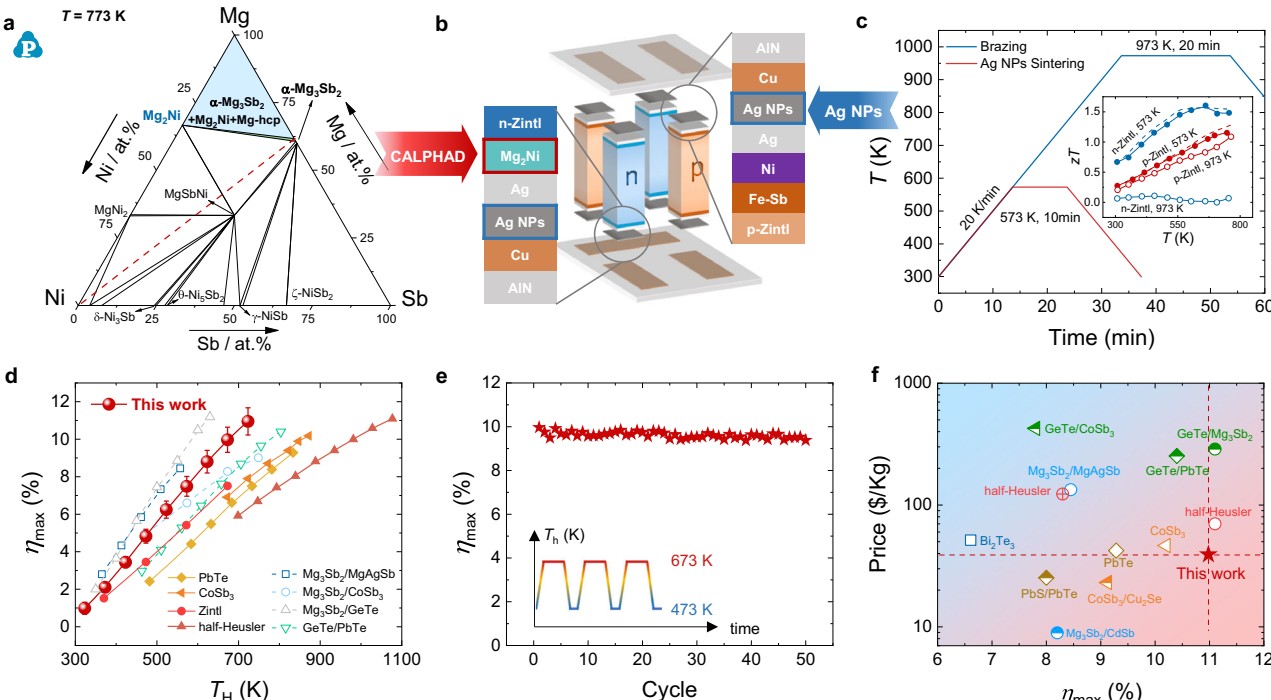

**Fig. 1 | Design, fabrication, and characterization of Zintl phase thermoelectric device. a** The calculated isothermal section at 773 K of the Mg-Ni-Sb ternary phase diagram in this work. Background colors in (**a**) represent the target phase area. **b** Schematic of Zintl phase device structure design, including thermoelectric legs, barrier layers, solder layers (Ag NPs), electrode layers, and pre-circuited AlN ceramic plates. **c** Comparison of temperature curves between traditional brazing process (Ag-Cu-Zn, melting point ~923 K) and low-temperature Ag NPs sintering process (welding temperature ~573 K). Insets: $zT$ values for n-type and p-type Zintl

materials after different welding process treatments. **d** Maximum conversion efficiency ($\eta_{max}$) as a function of hot-side temperature for the two-pair device. Literature data from other single-stage devices[9,32–38]. Error bars represent the result of uncertainty. **e** $\eta_{max}$ of the Zintl phase device throughout thermal cycling between hot-side temperatures of 473 K and 673 K. Inset: schematic illustration of the thermal cycling over time. **f** Comparisons of the price and $\eta_{max}$ of different TE devices[9,32–40].

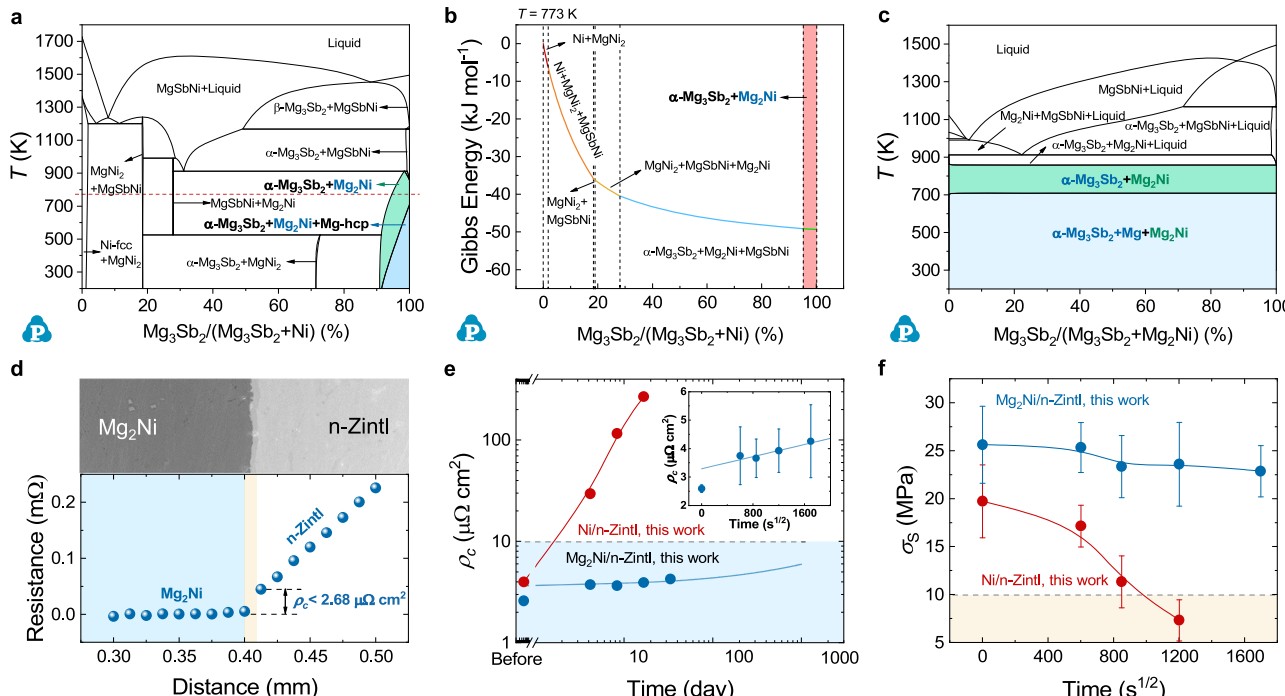

**Fig. 2 | Design and characterization of the Mg₂Ni/n-Zintl thermoelectric contact structures. a** Calculated vertical section of Ni-Mg₃Sb₂. **b** Calculated Gibbs energy for Ni-Mg₃Sb₂ at 773 K. **c** Calculated vertical section of Mg₂Ni-Mg₃Sb₂. Background colors in (**a–c**) represent the target phase area. **d** Surface morphology and measured contact resistivity ($\rho_c$) of Mg₂Ni/n-Zintl junction. Background colors in (**d**) represent the different material layers. **e** The fitting curve of the interfacial resistivity of Mg₂Ni junctions and the change of $\rho_c$ of Ni junctions aging at 673 K. Inset: Test values of $\rho_c$ of Mg₂Ni junctions during 0–800 H. **f** Plots of $\sigma_s$ versus the root mean square of the aging time at 673 K, showing linear relationships. Background colors in (**e**) and (**f**) represent the good range of $\rho_c$ and the unusable range of $\sigma_s$, respectively. Marks and error bars in (**e**) and (**f**) represent the mean value across three repetitions ($n = 3$) and standard deviation across five repetitions ($n = 5$), respectively.

welding effect and found that the TE properties degraded dramatically, especially in n-type samples (Inset in Fig. 1c). Compared with traditional brazing, Ag NPs sintering could be accomplished at 573 K for 10 min[24]. A similar simulation was performed and the results are presented in the inset of Fig. 1c, showing the unchanged properties at this temperature. Therefore, in order to ensure the thermoelectric properties of the material, it is necessary to avoid excessive welding temperature during the preparation of the Zintl device.

Finally, combined with the topologic structure design, a full Zintl phase device was fabricated. A record high conversion efficiency of ~11% in the single-stage Zintl phase device was achieved (Fig. 1d), which is ~33% higher than the state-of-the-art full Zintl phase device[32] (at $T_h$ = 673 K) and close to the other single-stage devices[9,32–38]. The thermodynamic stability between TE materials and barrier materials ensures the excellent long-term stability of the devices, which is verified by stable power output and conversion efficiency after 50 thermal cycles between hot-side temperatures of 473 K and 673 K (Fig. 1e). In addition, the Zintl phase device possesses low-cost price advantages[9,32–40] (Fig. 1f).

Herein, we propose an efficient screening strategy guided by diffusion doping effects and the CALPHAD method. A single pure metal cannot meet the requirements of the Mg₃Sb₂ material for the contact layer, which has been confirmed by experiments[41–43]. Since the diffusion of elements is inevitable, we need to follow the doping matching principle to select inert doping elements (such as Ni, Nb, Co, and Fe, etc. Supplementary Fig. 3). Mg is a necessary element to ensure the n-type conduction of Mg₃Sb₂-based materials, and can improve the CTE of the alloys to match with TE materials[44]. Querying the existing binary phase diagrams of Mg alloys, we choose Mg-Ni as a candidate. Based on the thermodynamic data of Mg-Ni[45–47], Mg-Sb[48], and Ni-Sb[49] binary phase diagram, we calculated the ternary phase diagram of Mg-Ni-Sb with the help of Pandat software (Supplementary Fig. 4). The

establishment of the multi-component phase diagram is able to predict the diffusion-reaction transformations between the two interfaces[50–52], and quickly screen out thermodynamically stable components among many multi-component phases. Taking the Ni-Mg₃Sb₂ connection interface as an example, we can make a vertical section along the connection line between Ni and Mg₃Sb₂ in the ternary phase diagram to check out the transformation at the interface. As shown in Fig. 2a, we obtained the equilibrium phase diagram of Ni-Mg₃Sb₂ at different ratios and temperatures. When the temperature is set as 773 K (the red dotted line in Fig. 2a), with the increase of Mg₃Sb₂ ratio, the predictable interfacial reaction structure may appear as Ni, Ni+MgNi₂, MgNi₂ + MgSbNi, MgSbNi+Mg₂Ni, and Mg₃Sb₂ + Mg₂Ni. Obviously, if the metal Ni is selected as the contact material, then the contact layer will undergo complex reactions. Besides, it should be noted that the mass balance line (the red dotted line in Figs. 1a, 2a) is not the diffusion path, but it can predict the interface structure to a certain extent. And the calculated Gibbs energy of the Mg₃Sb₂ + Mg₂Ni system is the lowest among all the composition systems, which means that this system is thermodynamically stable (Fig. 2b). The longitudinal profile of Mg₂Ni-Mg₃Sb₂ also proves that no chemical reaction occurs in the two-phase region regardless of the composition ratio from room temperature to 873 K, except for the solid solution and precipitation of Mg (Fig. 2c and Supplementary Fig. 5). These works are also carried out in Mg-Ni-Bi system[53,54], and we got the same conclusion (Supplementary Fig. 6). With the powerful thermodynamic calculation capability of CALPHAD, we can quickly screen out Mg₂Ni materials that are in thermodynamic equilibrium with Mg₃(Sb, Bi)₂, thus ensuring the thermal stability of the connection interface.

The Mg₂Ni/n-Zintl joints were then fabricated, and the interfacial structure evolution under accelerated aging time was systematically

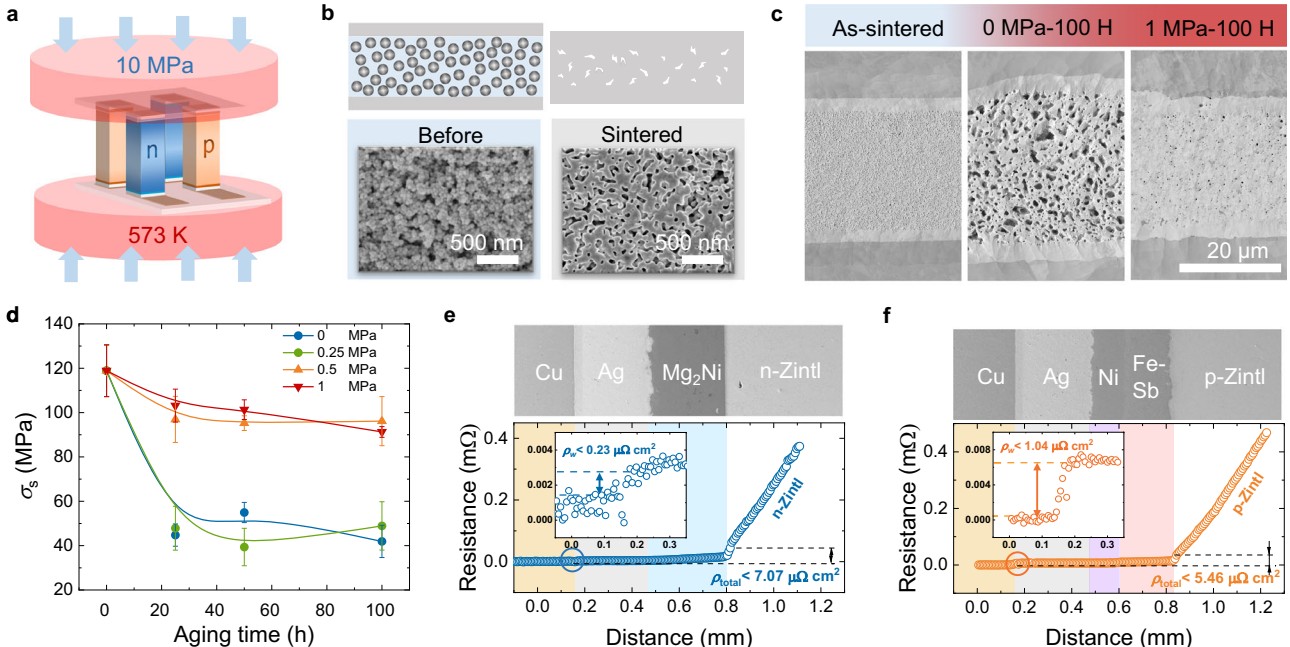

**Fig. 3 | Characterization of the nano-silver sintered connection layer.**
**a** Schematic diagram showing the assembly of the thermoelectric device by Ag NP sintered bonding. **b** SEM surface morphology and schematic diagram of Ag NP before and after sintering. **c** SEM cross-sectional morphologies of Ag NP joints aging at different durations and pressure, and (**d**) corresponding changes in $\sigma_s$. Marks and error bars represent the mean value and standard deviation across five repetitions ($n = 5$). Measured contact resistivity ($\rho_c$) of joints after sintering by Ag NP for the (**e**) n-type and (**f**) p-type junctions and corresponding to SEM cross-sectional morphology. Background colors in (**e**) and (**f**) represent the different material layers. Insets to (**e**) and (**f**): welding contact resistivity ($\rho_w$) of the Ag NP sintered interface.

investigated. According to elemental mapping results and compositional profiles (Fig. 2d and Supplementary Fig. 7), a thinner interlayer composed of Mg-Sb(Bi)-Ni exists at the interface of the prepared junction. Excess Ni observed in the $Mg_2Ni$ layer (Supplementary Fig. 8), considering the Mg volatilization during sintering, which caused the interface composition to partially deviate from the $Mg_2Ni$-$Mg_3Sb_2$ two-phase region, leading to the formation of the Mg-Sb(Bi)-Ni phase (corresponding phase diagram shown in Supplementary Fig. 9, and chemical compositions of interlayer list in Table S1). After 100 h of aging, the excess Ni near the interlayer interface was consumed. The interface entered a local thermodynamic equilibrium state, and no obvious change occurred in the subsequent 700-hour aging (Supplementary Fig. 10). Based on the experimental thickness of the intermediate layer during the aging process and the kinetic equation, the kinetic parameters and growth curves of the $Mg_2Ni$ joint at the aging temperature of 673 K were obtained (Supplementary Fig. 11). Since the degree of reaction is much weaker than that in direct contact with Ni, the initial contact resistivity of the $Mg_2Ni$ joint is only $2.5\,\mu\Omega\,cm^2$ (lower penal in Fig. 2d), which is also better than other reported joints so far[38,41,42,55]. Without considering other factors, the contact resistivity ($\rho_c$) is composed of interface electrical loss and intermediate layer loss, where the former is approximately constant and the latter is linear with the thickness of the intermediate layer[33]. If the thickness is calculated by diffusion theory, the change of contact resistance over time can be predicted (Fig. 2e), which shows that the contact resistivity is lower than $10\,\mu\Omega\,cm^2$ after 1000 days of aging. In contrast, the thickness of the intermediate layer in the Ni/n-Zintl joint rapidly increased to ~25 μm after aging for 400 h (Supplementary Figs. 12, 13). Moreover, due to the difference in the thermal expansion coefficient between Ni and n-Zintl, holes and cracks appear in TE materials near the interface. And the contact resistance of the Ni joint is greater than that of the $Mg_2Ni$ joint and increases to $>200\,\mu\Omega\,cm^2$ in 400 h (upper penal in Fig. 2e). Benefiting from the CTE matching of $Mg_2Ni$ with n-Zintl (Supplementary Fig. 14) and the defect-free dense interface, the $Mg_2Ni$

joint has satisfactory shear strength ($\sigma_s$) of ~25 MPa (Fig. 2f). And due to the thermodynamic stability of the designed interface structure, the decay of $\sigma_s$ is very slow in aging process, while the $\sigma_s$ of Ni/n-Zintl joint decays to less than 10 MPa. More importantly, the matching contact materials were developed in p-type $Mg_3Sb_2$ and PbTe materials through the CALPHAD method in this work (Supplementary Figs. 15, 16)[56–60], which proved the applicability of this screening strategy.

Since higher $zT$s tend to be achieved at elevated temperatures, a larger temperature difference for the same average $zT$ represents a higher efficiency. The pursuit of higher $zT$ and efficiency has brought the service temperature of TE materials close to the temperature limit of their thermal stability, which leads to an extremely narrow soldering temperature range for device fabrication[61–63]. The use of a conventional filler inevitably results in a low working temperature (unfavorable for the output power of the TE device) or a high welding temperature (unfavorable for the thermal stability of the TE material). In recent reports, the conductive filling paste was frequently used due to its simple application process and short-term high-temperature tolerance[6,64]. However, the decline of conductivity at high temperatures and the complete sacrifice of strength limits the application prospects of the prepared TE devices. Low-temperature connection technology can realize the connection at a temperature lower than the service temperature to avoid performance loss. Our previous work proved that Ag NPs as welding fillers could realize the preparation and service of low-, medium- and even high-temperature thermoelectric devices[24]. The thermal stability and reliability of low-temperature sintered nano-silver at high temperatures are concerns in the field of thermoelectricity, which has not been fully studied yet.

Due to the increase in specific surface energy caused by the nanonization of particles, nano-silver can be sintered at a lower temperature and pressure[65]. Figure 3a presents the schematic diagram of the nano-silver sintering process using the sintering temperature of 573 K and pressure of 10 MPa in this work. The morphology of Ag NPs before and after sintering has been characterized, and the results are

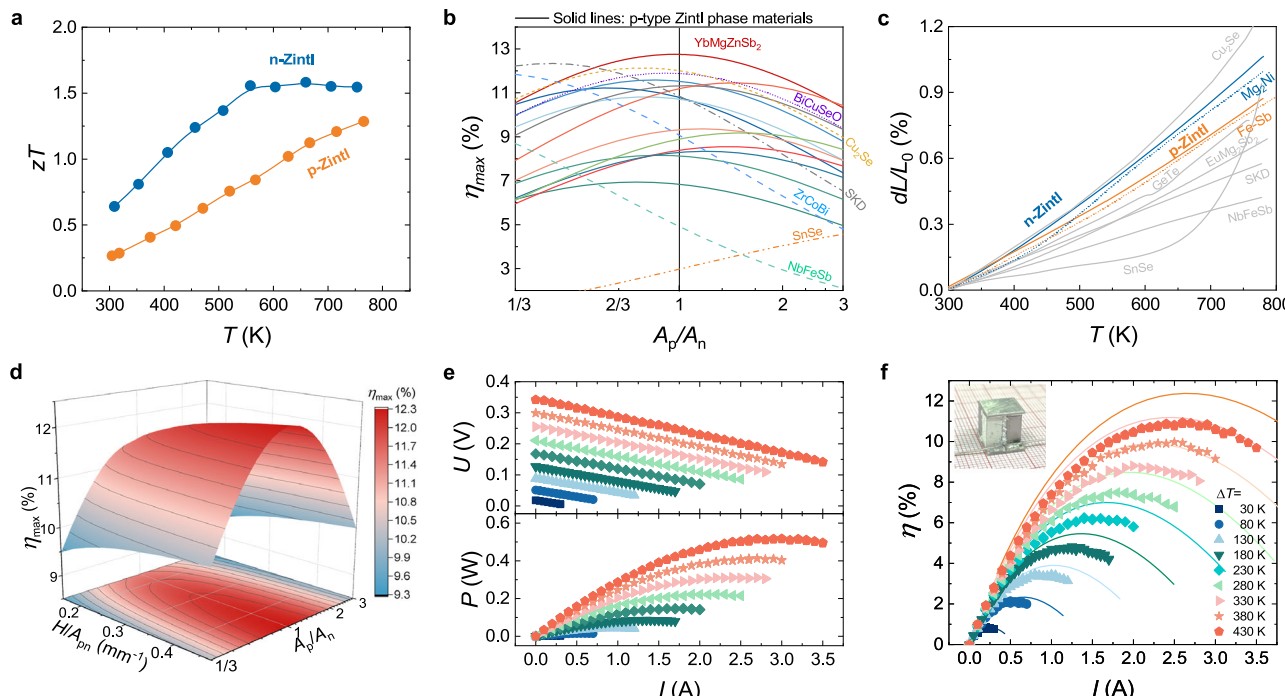

**Fig. 4 | Performance of the Zintl phase device. a** The $zT$ values for n-type $Mg_{3.15}Co_{0.05}SbBi_{0.99}Se_{0.01}$ and p-type $Yb_{0.9}Mg_{0.9}Zn_{1.198}Ag_{0.002}Sb_2$. **b** Finite-element-simulated maximum efficiency ($\eta_{max}$) as a function of the cross-sectional area ratio of the p- and n-type legs ($A_p/A_n$) Solid lines: n-type $Mg_{3.15}Co_{0.05}SbBi_{0.99}Se_{0.01}$ with p-type Zintl phase materials[69–79]. Dashed lines: n-type $Mg_{3.15}Co_{0.05}SbBi_{0.99}Se_{0.01}$ with other p-type materials[80–85]. **c** Temperature-dependent expansion behaviors of n-type $Mg_{3.15}Co_{0.05}SbBi_{0.99}Se_{0.01}$ compared with different p-type materials[90] and their connection materials in the range of 293-773 K. **d** Simulated $\eta_{max}$ as a function of the cross-sectional area ratio of the p- and n-type legs ($A_p/A_n$) and the ratio of height to the total cross-sectional area of the legs ($H/A_{pn}$). **e** Measured voltage ($U$) and output power ($P$) as a function of current ($I$) at different temperatures. **f** The experimental and calculated conversion efficiency as a function of $I$ at different hot-side temperatures. Insets: optical images of the Zintl phase device.

shown in Fig. 3b. The Ag NPs have an approximately spherical shape with a diameter of about 50 nm and are covered with an organic layer to prevent oxidation and self-adhesive connections. After sintering, the protection layer evaporates, and the particles are connected to each other to form a network structure, and the re-melting temperature returns to the melting point of silver due to the increase in size[24]. Considering the higher service temperature of the TE device than other normal electronic devices, the mechanical reliability of solder joints during aging has been studied. The nano-silver network structure of as-sintered joints is uniformly distributed and well connected at the interface, as shown in Fig. 3c. The average $\sigma_s$ of the as-sintered joint is up to 117 MPa (Fig. 3d), which is higher than that of many traditional welded joints[66]. However, during the long-aging process at 773 K, the sintered network structure grows, so the uniformly distributed micropores accumulate into larger pores[67,68], which causes the $\sigma_s$ of the joint to drop rapidly to 45 MPa within 25 h and maintain at 40–65 MPa within the subsequent 800-hour aging process. Thus we imposed different pressures during the aging process. The samples undergo the process of secondary sintering with the $\sigma_s$ of the joint maintaining above 90 MPa during the 100-hour aging process under the pressure of 0.5–1 MPa. Due to the densification of the nano-silver structure accompanied by lateral shrinkage, there are residual pores in part of the joint structure (Fig. 3c and Supplementary Fig. 17), the $\sigma_s$ of the joint cannot exceed the strength before aging, but it is completely sufficient to meet the service requirements of TE devices. In addition, the average contact resistivity of the Ag NP bonding interface is about 1 μΩ cm² (Insets in Fig. 3e, f), which is negligible compared to the internal resistance of the thermoelectric legs.

A highly efficient and reliable device requires n- and p-type materials with high TE performance and matched physical properties to reduce device residual thermal stress. Through a series of material optimization, our previous work has achieved high TE properties in both n- and p-type Zintls, i.e., $Mg_{3.15}Co_{0.05}SbBi_{0.99}Se_{0.01}$ (n-Zintl) and $Yb_{0.9}Mg_{0.9}Zn_{1.198}Ag_{0.002}Sb_2$ (p-Zintl)[5,31]. Temperature-dependent electrical and thermal transport properties of the materials are measured parallel and perpendicular to the hot-pressing direction, both of which are almost similar (Fig. 4a and Supplementary Figs. 1, 2).

A sandwich structure of Ag/Mg$_2$Ni/n-Zintl enables low total interfacial resistivity ($\rho_{total}$) of 7.1 μΩ cm² (Fig. 3e and Supplementary Figs. 18). Combined with Ag NPs sintering, we successfully prepared an n-Zintl single-leg device that can safely serve at high temperatures, and the results are shown in Supplementary Fig. 19 (As for comparison, the simulation results are provided in Supplementary Fig. 20). The single-leg device achieves a record efficiency of 13.3% and an output power density of 1.2 W cm⁻² at a temperature difference of 430 K. Under a high current density of 16 A cm⁻², the efficiency is basically unchanged after aging for 100 h at $T_h$ = 723 K and a $\Delta T$ = 430 K (attenuation rate is less than 5%) (Supplementary Fig. 21), indicating the high thermal stability of Mg$_2$Ni and sintered Ag NPs joints. The $zT$ value and theoretical efficiency of $Yb_{0.9}Mg_{0.9}Zn_{1.198}Ag_{0.002}Sb_2$ outperform the previously reported 1-2-2-type Zintl materials and are comparable to other p-type materials in the tested temperature range (Supplementary Fig. 22)[69–85]. The sandwich structure of Ag/Ni/Fe-Sb[86] is used to prepare the p-Zintl single-leg device, resulting in low $\rho_{total}$ values of 5.5 μΩ cm² (Fig. 3f and Supplementary Figs. 23). An efficiency of ~9.17% (theoretical efficiency is 11%) for this p-Zintl single-leg device at $T_h$ = 723 K and a $\Delta T$ = 430 K has been realized (Supplementary Fig. 24, 25). As a comparison, 11 types of state-of-the-art p-type Zintl-phase materials and 6 types of other materials[69–85] were selected, and the theoretical efficiency of devices constructed using them along with n-Zintl was calculated (Fig. 4b). Due to the difference in electrical and thermal properties, TE pairs composed of different n- and p-type TE materials will achieve the optimal conversion efficiency at different cross-sectional area ratios. $A_p/A_n$ = 1 is desirable since the heat transfer and thermal stress are more uniform under this

condition, and the filling fraction of TE legs can be increased. Finite element simulations were employed to obtain the optimal cross-sectional area ratio ($A_p/A_n$) of the p- and n-type legs for maximum conversion efficiency ($\eta_{max}$). According to the simulation results, p-type Zintl phase materials achieve the highest efficiency near $A_p/A_n = 1$ due to similar thermal and electrical properties, but the device efficiency is relatively low. The $A_p/A_n$ of other advanced TE materials is different from unity due to the difference in physical properties, but high $zT$s increase the overall efficiency of the device. Among them, p-Zintl is one of the few materials that simultaneously achieves $A_p/A_n$ close to 1 and has a high device efficiency, reaching ~13%. Figure 4c shows the thermal expansion properties of the n-Zintl and other p-type materials, where the thermal expansion performance of p-Zintl well matches that of n-Zintl.

In the geometry design of the thermoelectric device, we include the interfacial resistance and consider the effect of height. For a fixed total cross-sectional area ($A_{pn}$), the output performance and conversion efficiency of the device drop rapidly as $H$ decreases (Fig. 4d and Supplementary Fig. 26). Since the thermocouple is placed at the interface between the device and the heater or heat sink (the temperature at both ends of device is fixed during simulation), as $H$ decreases, the influence of the interface thermal resistance increases, and the actual temperature difference between the two ends of the TE legs decreases. In order to verify the performance and reliability of the device, a two-couple Zintl phase device with $A_p/A_n = 1$ was prepared and characterized. At the cold-side temperature of 293 K, the current ($I$), output voltage ($U$), $P$, and $\eta_{max}$ of this device were measured under different $\Delta T$ (Figs. 4e, f). A record $\eta_{max}$ of ~11% was realized when $T_h$ reached 723 K, close to the predicted value of ~12% (Supplementary Figs. 27, 28). The maximum power density ($\omega_{max}$) of the device was 0.51 W cm$^{-2}$, close to the calculated value of 0.54 W cm$^{-2}$. After 50 thermal cycles between hot-side temperatures of 473 K and 673 K, the internal resistance and output performance of the Zintl device remain stable, i.e., the attenuation rate is less than 8% (Fig. 1e and Supplementary Fig. 29). The high thermal stability of the device can be partly attributed to the matching physical properties of p- and n-type Zintl phase materials.

To sum up, the computational phase diagram method was successfully applied to efficiently screen thermodynamically stable contact materials. It has been demonstrated that Mg$_2$Ni/Mg$_{3.15}$Co$_{0.05}$SbBi$_{0.99}$Se$_{0.01}$ joints have excellent contact electrical properties and thermal stability. The low-temperature Ag NPs sintering substantially reduces the welding temperature to ensure the integrity of the interface and the high thermoelectric performance of the materials, and the proven high-temperature service reliability ensures the stability of the thermoelectric devices. The single-leg and two-couple devices with Mg$_2$Ni/Mg$_{3.15}$Co$_{0.05}$SbBi$_{0.99}$Se$_{0.01}$ junctions achieve high $\eta$ of ~13.3% and ~11%, respectively, with good stability and thermal cycling performance. Our work not only proves the potential of Zintl thermoelectric materials and devices, but also provides a feasible path from contact design to device connection, thus greatly promoting the development of advanced thermoelectric power generation devices with high efficiency and remarkable reliability.

## Methods

**Preparation and measurement of joints and devices.** High-purity raw materials were directly weighed according to the nominal composition Mg$_{3.15}$Co$_{0.05}$SbBi$_{0.99}$Se$_{0.01}$ (n-Zintl), loaded into a ball-milling jaw in a glovebox, and finally subjected to the milling process for 10 h (SPEX SamplePrep 8000 Mixer Mill). The p-type Yb$_{0.9}$Mg$_{0.9}$Zn$_{1.198}$Ag$_{0.002}$Sb$_2$ (p-Zintl) sample was synthesized by ball-milling for 5 h[31]. The p-type Mg$_{2.475}$Zn$_{0.5}$Na$_{0.0125}$Sb$_2$ materials were obtained by ball-milling the raw materials for 10 h[71]. The p-type Na$_{0.02}$Pb$_{0.98}$Te materials were obtained by melting at 1273 K for 6 h[24,70]. The contact materials Mg$_2$Ni, Fe-Sb and Fe-Te alloys were also obtained by ball-milling the raw materials for 10 h. The raw powders of Mg$_2$Ni were sintered using the SPS method

under 40 MPa at 773 K for 10 min. Then, the sintered bulk was ball milled for 6 min to get the final powders.

After sintering the n-Zintl powders at 1023 K under 50 MPa for 2 min, the bulk sample was polished and sintered with Mg$_2$Ni powders and Ag powders at 773 K under 40 MPa for 10 min to obtain the n-Zintl junctions. The p-Zintl powders were sintered with Fe-Sb powders, Ni powders, and Ag powders at 923 K under 40 MPa for 5 min. The p-type Mg$_3$Sb$_2$/Fe-Sb junctions were prepared in a similar way. The p-type PbTe/Fe-Te junctions were obtained by hot-pressing at 773 K for 5 min under a pressure of 50 MPa. For the aging test, there are at least three parallel specimens for each aging time.

The sintered junctions were cut into about $3.6 \times 3.6 \times 7$ mm$^3$. After grinding and polishing, a protective coating of boron nitride from a commercial source was sprayed on the four sides of the n-Zintl thermoelectric legs, while the hot and cold ends were shielded. Then the hot and cold sides of the TE legs were sintered onto Ag electrode foil or Ag-plated copper metalized ceramic substrate using Ag NP paste at 573 K for 10 min under 10 MPa. The same sintering process was used to prepare welding joints on Ag-plated copper sheets for the shear strength test. The overall size of the 2-pair device was about $10 \times 10 \times 9$ mm$^3$. Copper wires were soldered onto the cold side of Cu electrodes to measure the current and voltage. Aerogel slurry was filled within the thermoelectric legs to reduce heat loss due to convection and radiation. The measurements of the device were conducted in a homemade testing system[41].

**Characterization methods.** A commercial system (ULVAC ZEM-3) was used to characterize the electrical conductivity ($\sigma$) and Seebeck coefficient ($S$). Thermal conductivity ($\kappa$) was calculated by $\kappa = d \times C_p \times D$, where $d$ is the density measured by the Archimedean method, $C_p$ is the specific heat capacity measured by differential scanning calorimetry (Netzsch DSC 404 C), and $D$ is the thermal diffusivity measured by laser flash method (Netzsch LFA457). The coefficient of thermal expansion was determined by the thermal mechanical analyzer (TMA 402, Netzsch). Scanning electron microscopy (SEM, Phenom Pro) and an attached energy dispersive spectrometer were used to analyze the microstructure and phase composition. A bonding testing instrument (TRY, MFM-1200) was used to characterize the shear strength with a shear height of 100 μm and shear speed of 200 μm s$^{-1}$. A homemade four-probe measurement system[41] was used to measure the interfacial resistivity of the junctions with an alternating current of 0.1 A. The geometric optimization of thermoelectric devices was simulated and analyzed by finite element analysis software COMSOL Multi-physics[24,87]. The simulation boundary conditions were determined according to Supplementary Table 2.

**CALPHAD modeling.** We take the Mg-Ni-Sb as an example to explain the CALPHAD process in detail. Thermodynamic modeling of the Mg-Ni-Sb system was based on data from Mg-Ni[45–47], Mg-Sb[48], and Ni-Sb[49] binary subsystems, and the corresponding binary phase diagrams are presented in Supplementary Fig. 2. All phases involved in Mg-Ni-Sb can be divided into the solution phases and the intermetallic phases. The Gibbs energy of pure elements at stable element reference (SER, 298.15 K and 10$^5$ Pa) were taken from the SGTE (Scientific Group Thermodata Europe) database[88].

The solution phases based on the elements of Mg, Ni, Sb, and the liquid are described with a substitutional solution model and associate solution model, whose (phase $\varphi$) molar Gibbs energy could be defined as:

$$G_m^\varphi = \sum_i x_i \, ^0G_i^\varphi + RT \sum_i x_i \ln(x_i) + \,^E G_m^\varphi \tag{1}$$

where $x_i$ and $^0G_i$ respectively denotes the mole fraction and the ideal molar Gibbs energy of a phase for the element $i$, $R$ the gas constant, $T$ the absolute temperature, $^E G_m$ the excess molar Gibbs energy that is

expressed as Redlich-Kister formula:

$$^E G_m^\varphi = \sum_i \sum_{j>i} x_i x_j \sum_{v=0}^{n} (x_i - x_j)^v \cdot {}^v L_{i,j}^\varphi + \sum_i \sum_{j>i} \sum_{k>j} x_i x_j x_k (x_i {}^0 L_{i,j,k}^\varphi \\ + x_j {}^1 L_{i,j,k}^\varphi + x_k {}^2 L_{i,j,k}^\varphi)$$

(2)

where $L_{i,j}$ is the binary interaction parameters of the species $i$ and $j$, and $^v L_{i,j,k}$ is the ternary interaction parameters of the species $i$, $j$, and $k$ ($v = 0, 1, 2$ and $i, j, k = $ Mg, Ni, Sb, $Mg_3Sb_2$). The binary and the ternary interaction parameters take the following form:

$$^v L_m^\varphi = a + b \cdot T + c \cdot T \cdot \ln(T) + d \cdot T^2 + e \cdot T^{-1} + f \cdot T^3 + g \cdot T^7 + h \cdot T^{-9}$$

(3)

where $a, b, c, d, e, f, g$ and $h$ are the coefficients to be optimized. In most cases, only the first one or two terms of the above equation are used.

There are 10 intermetallic phases existing in the Mg-Ni-Sb system, in which the $Mg_2Ni$, $\zeta$-$NiSb_2$, and MgNiSb were modeled as the stoichiometric compounds. And the solubility of the other element in $\alpha$-$Mg_3Sb_2$, $\beta$-$Mg_3Sb_2$, $MgNi_2$, $\delta$-$Ni_3Sb$, $\beta$-$Ni_3Sb$, $\theta$-$Ni_5Sb_2$, and $\gamma$-NiSb was considered. All the intermetallic phases are treated as a sublattice model and the Gibbs energy is described based on the Compound Energy Formalism (CEF) proposed by Mats Hillert[89].

The crystallographic data of all the phases in the Mg-Ni-Sb system are listed in Table S3 and the optimized parameters are listed in Table S4. Similar CALPHAD work has also been carried out for the Mg-Ni-Bi system, Mg-Sb-Fe system, and Pb-Te-Fe system, and the corresponding information is listed in Table S5 to Table S8. All calculations in this work were performed by Pandat software.

**Uncertainty analysis.** The uncertainty of the temperature difference ($\Delta T_{Cu}$), heat flow ($Q$), output power ($P$), and efficiency ($\eta$) was calculated by standard error analysis and propagation method as following formulas:

$$\delta(\Delta T_{Cu}) = \sqrt{[T_{Cu1} \times \delta(T_{Cu1})]^2 + [T_{Cu2} \times \delta(T_{Cu2})]^2} / (T_{Cu1} - T_{Cu2})$$

(4)

$$\delta(Q) = \sqrt{\delta(\Delta T_{Cu})^2 + \delta(\kappa_{Cu})^2 + \delta(A_{Cu})^2 + \delta(L_{Cu})^2}$$

(5)

$$\delta(P) = \sqrt{\delta(I)^2 + \delta(U)^2}$$

(6)

$$\delta(P+Q) = \sqrt{[P \times \delta(P)]^2 + [Q \times \delta(Q)]^2} / (P+Q)$$

(7)

$$\delta(\eta) = \sqrt{\delta(P)^2 + \delta(P+Q)^2}$$

(8)

$$\delta(w) = \sqrt{\delta(P)^2 + \delta(A_{Cu})^2}$$

(9)

The uncertainty of $\kappa_{Cu}$, $A_{Cu}$, $L_{Cu}$ (8 mm ± 0.1 mm), $I$, $U$, and $T$ are 7%, 0.5%, 1.25%, 1%, 1%, and ±0.01 K, respectively. Table R9 lists the estimated uncertainty of $\Delta T_{Cu}$, $Q$, and $\eta$ measurements for the Zintl-based devices.

## Data availability

All data are available in the main text or the supplementary materials. The data that support the findings of this study are available from the corresponding author upon reasonable request.

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

## Acknowledgements

This work was supported by the National Key Research and Development Program of China (2023YFB3809400), the Shenzhen Science and Technology Program (KQTD20200820113045081). Q.Z. acknowledges financial support from the National Natural Science Foundation of China (52172194, 51971081), the Natural Science Foundation for Distinguished Young Scholars of Guangdong Province of China (2020B1515020023), and the Shenzhen Science and Technology Program (RCJC20210609103733073). X.F.L. acknowledges financial support from the National Natural Science Foundation of China (52302232), the China Postdoctoral Science Foundation (2022M720941). J.M. acknowledges financial support from the National Natural Science Foundation of China (52101248), Shenzhen fundamental research projects (JCYJ20210324132808020).

## Author contributions

Q.Z. and L.Y. designed this work. L.Y., X.F.L., C.C., and Z.W.Z. synthesized the samples and conducted the transport property measurements. X.F.L. conducted the calculation of the phase diagram. L.Y., X.B., and J.X.C. conducted the experimental study on device fabrication and characterization. Q.Z., L.Y., X.F.L., J.M., F.C., and X.J.L. discussed the results. L.Y., X.F.L., Q.Z., J.M., and F.C. wrote this manuscript and all authors edited it.

## Competing interests

The authors declare no competing interests.
