## [Peer Review File · Nature Communications]

CALPHAD accelerated design of advanced full-Zintl thermoelectric deviceREVIEWER COMMENTS

Reviewer #1 (Remarks to the Author):

Because of the incorporation of chemically active elements, the device development of Zintl-phase compounds is more difficult. The reports of thermoelectric devices completely made of Zintl-phase compounds are even rarer. In this work, the authors innovatively used CALPHAD as a tool to seek suitable barrier layer interface materials. Experimental evidence showed that Mg₂Ni served as a barrier layer with high stability. In addition, they combined nano-silver sintering technology to provide a low-temperature connection method for Zintl devices. The efficiency of the full Zintl phase module reached 11%, and there was no significant decay after 50 cycles. This work demonstrates that CALPHAD has a considerable advantage in designing thermoelectric device interfaces, and can provide guidance for other thermoelectric systems. The novelty of this work is distinctive, the discussion is clear, and I suggest that it be published after minor revisions. Here are the comments:

1. Why did the authors ultimately select Ni when conducting CALPHAD, rather than other elements such as Co, Fe, Nb, etc.?
2. How was the Mg₂Ni mentioned in the paper prepared? Is there an XRD of the powder?
3. From Fig. S1, the Seebeck coefficient exhibits some anisotropy. According to earlier reports, polycrystalline Mg₃Sb₂ samples do not exhibit obvious anisotropy. It is suggested that the authors confirm this test result.
4. The measurement uncertainties of efficiency should be provided and the corresponding error bars need to be added in Fig. 1d and S24.
5. In Fig. 4f, S16 and S21, a white coating is added to the surface of n-type legs but not p-type. The authors should add clarification.

Reviewer #2 (Remarks to the Author):

This study provides a solid groundwork for enhancing the performance of devices based on Zintl compound materials. Distinguishing itself from previous research, the primary innovation of this work lies in the choice of Mg₂Ni as the contact layer for Mg₃SbBi. This selection effectively reduces interface contact resistance, leading to an improvement in device efficiency. Nonetheless, certain limitations are also present in this study.

- 1) The calculation of phase diagram method proposed by the authors is exclusively employed to

determine the Mg₂Ni contact layer in n-type Mg₃Sb₂. The range of contact layer materials solely based on Mg-Sb-Ni phase diagram data in this study is confined. Can this method be broadly applicable in identifying the most suitable contact layer across various TE materials? This necessitates an extensive phase diagram database. In addition, the authors solely investigate n-type Mg₃SbBi material and Mg₂Ni contact layer, potentially rendering the paper's title somewhat overly exaggerated.

2) The method used in this work calculates the Mg-Sb-Ni ternary phase diagram and related Gibbs energy. However, the experimental n-type materials are 50% Mg₃Sb₂- 50% Mg₃Bi₂ alloys. In this case, does this calculation guide us to conclude that Mg₂Ni is suitable for both Mg₃Sb₂ and Mg₃Sb₂-Mg₃Bi₂ alloys?

3) The study involves a comparison of contact resistivity and thermal stability between the Ni/Mg₃Sb₂ junction and the Mg₂Ni/Mg₃SbBi junction. To establish a comprehensive comparison, it is advisable to include the results of Ni joint or the commonly-used Fe joint in Mg₃SbBi alloys.

4) From Supplementary Fig. 9, it can be seen that a reaction layer forms between the n-type Mg₃.15Co_{0.05}SbBi_{0.99}Se_{0.01} and Mg₂Ni, and the thickness of this reaction layer changes over time. What is the chemical composition of this reaction layer? And how does it affect the contact resistivity?

5) Certain literature related to Mg₃SbBi devices with high efficiency have not been included for comparison in this research (Energy Environ. Sci., 2021, 14, 6506–6513). Furthermore, the conversion efficiency reported in literature works such as Mg₃Sb₂/MgAgSb and Mg₃Sb₂/SKD is similar to that presented in this study within the temperature range of 300-600 K. The notably high efficiency achieved in this work is within the temperature range of 600-723 K. What is the primary reason for this?

Considering the criteria for innovation, it might not perfectly align with the publication standards of Nature Communications. I believe that addressing these issues will uphold the work's commendable level of quality.

Response to reviewers

Reviewer #1:

Because of the incorporation of chemically active elements, the device development of Zintl-phase compounds is more difficult. **The reports of thermoelectric devices completely made of Zintl-phase compounds are even rarer.** In this work, the authors innovatively used CALPHAD as a tool to seek suitable barrier layer interface materials. Experimental evidence showed that Mg_2Ni served as a barrier layer with high stability. In addition, they combined nano-silver sintering technology to provide a low-temperature connection method for Zintl devices. The efficiency of the full Zintl phase module reached 11%, and there was no significant decay after 50 cycles. **This work demonstrates that CALPHAD has a considerable advantage in designing thermoelectric device interfaces, and can provide guidance for other thermoelectric systems.** The novelty of this work is distinctive, the discussion is clear, and I suggest that it be published after minor revisions. Here are the comments:

Response: We thank the reviewer very much for the positive comments and appreciate the valuable suggestions that have greatly helped us to improve the manuscript.

Comments (1-1):

Why did the authors ultimately select Ni when conducting CALPHAD, rather than other elements such as Co, Fe, Nb, etc.?

Response: We understand the reviewer's concern about the principle of selecting elements in our screening strategy. As we stated in the manuscript, we follow the doping matching principle to select the inert or same doping elements (such as Ni, Nb, Co, and Fe, etc.). The coefficient of thermal expansion of these single metals is different from that of Mg_3SbBi alloy, as shown in Fig. R1. Mg is a necessary element to ensure the n-type conduction of Mg_3SbBi -based materials, and can improve the CTE of alloys. We then query the existing magnesium alloy binary phase diagram, as shown in Fig. R2 below. The binary phase diagrams show that there is neither solid solution nor compound formation between the Fe or Nb elements and Mg. The presence of Mg metal is not conducive to the preparation of the connection layer. As for Co and Ni elements, there is MgCo_2 compound in Mg-Co phase diagram, and Mg_2Ni and MgNi_2 compounds in the Mg-Ni phase diagram. The high content of Mg compound is the reason why we chose Ni for the next calculation of the multi-phase diagram, which is more likely to achieve the matching of thermal expansion coefficients. We also believe that Mg_2Ni is not the only suitable contact layer material and that our screening principles and CALPHAD method can continue to develop highly matched contact layer materials in other potential elements.

Fig. R1| Characterization of thermal expansion behavior of metals and alloys. Temperature-dependent expansion behaviors of Mg_{3.15}Co_{0.05}SbBi_{0.99}Se_{0.01}, Mg₂Ni, 304 stainless steel, Fe, Ni, Co and Nb in the range of 293-773 K: (a) dL/L_0 and (b) CTE.

Fig. R2| The reference Mg-X (Nb, Fe, Co, and Ni) binary phase diagrams. (a) Mg-Nb binary phase diagram, (b) Mg-Fe binary phase diagram, (c) Mg-Co binary phase diagram, and (d) Mg-Ni binary phase diagram.

Comments (1-2):

How was the Mg₂Ni mentioned in the paper prepared? Is there an XRD of the powder?

Response: We added the preparation process of Mg₂Ni in the method part. The contact material Mg₂Ni alloys were obtained by ball milling the raw materials (Ni powders, 99.99%; Mg chips, 99.99%) for 10 hours. The raw powders (Powder I) were sintered using the SPS method under 40 MPa at 773 K for 10 min. Then, the sintered bulk was ball milled for 6 min to get the final powders (Powder II). The XRD results show that the main components of sintered powders are Mg₂Ni and a small amount of Mg, which will completely volatilize during secondary sintering to help protect n-Zintl bulk (Fig. R3).

Fig. R3| XRD pattern of Mg₂Ni powders and bulk.

Comments (1-3):

From Fig. S1, the Seebeck coefficient exhibits some anisotropy. According to earlier reports, polycrystalline Mg₃Sb₂ samples do not exhibit obvious anisotropy. It is suggested that the authors confirm this test result.

Response: Thanks for your suggestion. We re-prepared sintered bulk with the thickness of 12 mm and tested the parallel and vertical thermoelectric properties, as shown in Fig. R4. As previously reported, there are slight differences in electrical and thermal properties, and their zT are the same. In general, polycrystalline Mg₃Sb₂ samples do not have obvious anisotropy.

Fig. R4| Temperature-dependent thermoelectric properties of the n-type $\text{Mg}_{3.15}\text{Co}_{0.05}\text{SbBi}_{0.99}\text{Se}_{0.01}$ materials. (a) Electrical resistivity, (b) Seebeck coefficient, (c) thermal conductivity, and (d) zT value as a function of temperature.

Comments (1-4):

The measurement uncertainties of efficiency should be provided and the corresponding error bars need to be added in Fig. 1d and S24.

Response: We appreciate the reviewer's valuable suggestions. We have added error bars for maximum conversion efficiency in Fig. 1d and S28 (as shown in Fig. R5) and discussed and provided data for the uncertainty analyses. Relevant revisions to the manuscript and Supplementary Materials are shown below.

Uncertainty analysis

An uncertainty analysis was conducted using the standard error analysis and propagation method. The measurement of κ_{Cu} has an uncertainty of 7% and the uncertainties of A_{Cu} , L_{Cu} ($8 \text{ mm} \pm 0.1 \text{ mm}$), I , U and T are 0.5%, 1.25%, 1%, 1%, and $\pm 0.01 \text{ K}$, respectively. The uncertainty of the temperature gradient (ΔT_{Cu}), heat flow (Q), output power (P), and efficiency (η) can be calculated using the following formulas:

$$\delta(\Delta T_{\text{Cu}}) = \sqrt{(T_{\text{Cu}1} \times \delta(T_{\text{Cu}1}))^2 + (T_{\text{Cu}2} \times \delta(T_{\text{Cu}2}))^2} / (T_{\text{Cu}1} - T_{\text{Cu}2}) \quad (\text{S1})$$

$$\delta(Q) = \sqrt{\delta(\Delta T_{\text{Cu}})^2 + \delta(\kappa)^2 + \delta(A_{\text{Cu}})^2 + \delta(L_{\text{Cu}})^2} \quad (\text{S2})$$

$$\delta(P) = \sqrt{\delta(I)^2 + \delta(U)^2} \quad (S3)$$

$$\delta(P + Q) = \sqrt{(P \times \delta(P))^2 + (Q \times \delta(Q))^2} / (P + Q) \quad (S4)$$

$$\delta(\eta) = \sqrt{\delta(P)^2 + \delta(P + Q)^2} \quad (S5)$$

$$\delta(\omega) = \sqrt{\delta(P)^2 + \delta(A_{Cu})^2} \quad (S6)$$

Table R1 lists the estimated uncertainty of the temperature gradient (ΔT_{Cu}), heat flow (Q), and conversion efficiency (η) measurements for the Zintl-based modules. The uncertainty of Q is about 32.85%-7.26%, decreasing with increasing ΔT_{Cu} . Output power of each module is determined by its current (I) and output voltage (V) and has an uncertainty of 1.41%. The uncertainty of η ranges from 32.55% to 6.62% and is dependent on the temperature gradient applied, with a larger temperature gradient helping to reduce the uncertainty.

Fig. R5| Characterization of the Zintl-based thermoelectric module. (a) Maximum conversion efficiency (η_{max}) as a function of hot-side temperature for the two-pair module. Literature data from other single-stage modules¹⁻⁸. (b) Comparison between measured η_{max} and simulated values under different temperature gradients.

Table R1| Estimated uncertainty of the ΔT_{Cu} , measured heat flow (Q), and conversion efficiency (η) of the Zintl-based module in this study.

ΔT (K)	ΔT_{Cu} (K)	$\delta(\Delta T_{Cu})$ (%)	$\delta(Q)$ (%)	$\delta(\eta)$ (%)
30	0.04	32.06	32.85	32.55
80	0.17	8.41	11.02	10.88
130	0.29	4.80	8.60	8.42
180	0.41	3.42	7.90	7.65
230	0.52	2.72	7.63	7.29
280	0.65	2.18	7.45	7.04

330	0.77	1.84	7.36	6.86
380	0.90	1.56	7.30	6.72
430	1.03	1.37	7.26	6.62

Comments (1-5):

In Fig. 4f, S16 and S21, a white coating is added to the surface of n-type legs but not p-type. The authors should add clarification.

Response: Thanks for your kind reminder. In order to inhibit the volatilization of Mg element under high temperature and high vacuum environment in n-type $\text{Mg}_3(\text{Sb,Bi})_2$ materials⁹, a protective coating of boron nitride was prepared on the surface of n-Zintl legs. And the description of the coating process is added in the Method section. The sintered bulks were cut into about $3.6 \times 3.6 \times 7 \text{ mm}^3$. After grinding and polishing, a protective coating of boron nitride from commercial source was sprayed on the four sides of the n-Zintl thermoelectric legs, while the hot and cold ends were shielded.

Reviewer #2:

This study provides a solid groundwork for enhancing the performance of devices based on Zintl compound materials. Distinguishing itself from previous research, the primary innovation of this work lies in the choice of Mg₂Ni as the contact layer for Mg₃SbBi. This selection effectively reduces interface contact resistance, leading to an improvement in device efficiency. Nonetheless, certain limitations are also present in this study.

Considering the criteria for innovation, it might not perfectly align with the publication standards of Nature Communications. **I believe that addressing these issues will uphold the work's commendable level of quality.**

Response: We thank the reviewer very much for the valuable suggestions that have greatly helped us to improve the manuscript. We are delighted that the improved manuscript addresses the reviewer's concerns.

Comments (2-1):

The calculation of phase diagram method proposed by the authors is exclusively employed to determine the Mg₂Ni contact layer in n-type Mg₃Sb₂. The range of contact layer materials solely based on Mg-Sb-Ni phase diagram data in this study is confined. Can this method be broadly applicable in identifying the most suitable contact layer across various TE materials? This necessitates an extensive phase diagram database. In addition, the authors solely investigate n-type Mg₃SbBi material and Mg₂Ni contact layer, potentially rendering the paper's title somewhat overly exaggerated.

Response: The calculation of phase diagram method is based on thermodynamic data of pure elements (the database constructed by Scientific Group Thermodata Europe, SGTE) and components (databases such as Fact Sage, Springer, *etc.*), which means that the method has a strong support of thermodynamic theoretical basis and experimental data. Through decades of experimental research on thermodynamic phase diagrams, researchers have established a complete thermodynamic database of pure elements and most experimental thermodynamic data of binary phase diagrams, which strongly supports the development of computational phase diagram technology.

We understand reviewer's concerns about the applicability of the CALPHAD method in other materials, so we performed phase diagram calculation and experimental verification in materials with different conductive types and different systems, as shown in Fig. R6 and R7. After screening the potential contact elements for p-type Mg₃Sb₂ and PbTe materials, we established the Mg-Sb-Fe and Pb-Te-Fe ternary phase diagrams based on CALPHAD method¹⁰⁻¹⁵. We can quickly screen out Fe, FeSb and FeSb₂ materials that are in thermodynamic equilibrium with Mg₃Sb₂, thus ensuring no chemical reaction at the interfaces. By changing the ratio of Fe to FeSb, we can find the 70Fe30Sb-alloy material with the CTE matching. The longitudinal profile of 70Fe30Sb-Mg₃Sb₂ also proves that no chemical reaction occurs in the three-phase region from room temperature to ~1000 K. The prepared 70Fe30Sb-Mg₃Sb₂ junction is

consistent with the predicted results of the phase diagram, and there is no chemical reaction at the interface. The contact resistivity is at a good level of $2.3 \mu\Omega \text{ cm}^2$, and there is no significant decay during the aging period of up to 800 hours. We also developed 70Fe30Te contact material for p-type PbTe materials and obtained ideal contact resistivity and thermal stability. The above results show that the CALPHAD method is a general and effective technique for screening contact materials of thermoelectric devices. We are also working on building a database specifically designed for barrier materials in thermoelectric materials.

Thanks for your valuable suggestion. To better match our main work, we have modified the manuscript's title to "CALPHAD accelerated design of advanced full-Zintl thermoelectric device".

Fig. R6| The calculated phase diagram and the experimental results. The isothermal sections of (a) Mg-Sb-Fe ternary phase diagram at 673 K and (d) Pb-Te-Fe ternary phase diagram at 773 K. Calculated vertical sections of (b) 70Fe30Sb-Mg₃Sb₂ and (e) 70Fe30Te-PbTe. Surface morphology and measured contact resistivity (ρ_c) of (c) 70Fe30Sb/Mg₃Sb₂ and (f) 70Fe30Te/PbTe junctions.

Fig. R7| Design and characterization of the thermoelectric contact structures. Thermal expansion behavior of (a) Fe-Sb alloys and p-type Mg_3Sb_2 material, and (d) Fe-Te alloys and p-type PbTe material. SEM images and EDS mapping results of (b) 70Fe30Sb/ Mg_3Sb_2 junctions and (e) 70Fe30Te/PbTe junctions after 800 hours aging. Corresponding changes in ρ_c of (c) 70Fe30Sb/ Mg_3Sb_2 junctions and (f) 70Fe30Te/PbTe junctions with different aging time.

Comments (2-2):

The method used in this work calculates the Mg-Sb-Ni ternary phase diagram and related Gibbs energy. However, the experimental n-type materials are 50% Mg_3Sb_2 -50% Mg_3Bi_2 alloys. In this case, does this calculation guide us to conclude that Mg_2Ni is suitable for both Mg_3Sb_2 and Mg_3Sb_2 - Mg_3Bi_2 alloys?

Response: We understand your concerns. In fact, based on the binary phase diagrams of Mg-Bi¹⁶, Mg-Ni¹⁷⁻¹⁹, and Bi-Ni²⁰, we have also calculated the ternary phase diagram of Mg-Bi-Ni (as shown in Fig. R8), which will be added to the SI file. At temperatures equal to or below 773 K, the Mg_3Bi_2 - Mg_2Ni - MgNiBi three-phase region can exist stably, indicating that Mg_2Ni does not chemically react with Mg_3Bi_2 under equilibrium conditions. Considering both the Mg-Bi-Ni and Mg-Sb-Ni ternary phase diagrams, we believe that Mg_2Ni can coexist stably with Mg_3Sb_2 and Mg_3Bi_2 . Therefore, Mg_2Ni alloy is suitable as a connecting layer material for Mg_3SbBi thermoelectric materials.

Fig. R8| The calculated phase diagram based on CALPHAD method. Calculated (a) Mg-Bi binary phase diagram, (b) Mg-Ni binary phase diagram, (c) Ni-Bi binary phase diagram, isothermal sections at (d) 300 K, (e) 723 K, and (f) 773 K.

Comments (2-3):

The study involves a comparison of contact resistivity and thermal stability between the Ni/Mg₃Sb₂ junction and the Mg₂Ni/Mg₃SbBi junction. To establish a comprehensive comparison, it is advisable to include the results of Ni joint or the commonly-used Fe joint in Mg₃SbBi alloys.

Response: We appreciate the reviewer’s valuable suggestions. In order to achieve the same preparation conditions, the Ni/Mg₃SbBi junction was prepared by the same two-step sintering method as the Mg₂Ni/Mg₃SbBi junction. After sintering at 773 K for 10 min, an intermediate layer with thickness of ~3 μm formed at the interface. And the thickness of intermediate layer rapidly increased to ~25 μm after aging for 400 h (Fig. R9 and R10). Moreover, due to the difference of thermal expansion coefficient of Ni and Mg₃SbBi, holes and cracks appear in Mg₃SbBi materials near the interface. With the increase of aging time, the number of holes increases and the crack propagation results in interface cracking. As a result, the contact resistivity of the Ni\Mg₃SbBi joint rapidly increases to more than 200 μΩ cm², and the connection strength decays to less than 10 MPa (Fig. R11).

Fig. R9| Characterizations of Ni/Mg_{3.15}Co_{0.05}SbBi_{0.99}Se_{0.01} junctions. SEM images and EDS mapping results of Ni/Mg_{3.15}Co_{0.05}SbBi_{0.99}Se_{0.01} junctions (a) as-prepared, and aging at 673 K for (b) 100 hours, (c) 200 hours, and (d) 400 hours.

Fig. R10| Characterizations of Ni/Mg_{3.15}Co_{0.05}SbBi_{0.99}Se_{0.01} junctions' reaction rate. SEM images and EDS line scanning results of Ni/Mg_{3.15}Co_{0.05}SbBi_{0.99}Se_{0.01} junctions (a) as-prepared, and aging at 673 K for (b) 400 hours.

Fig. R11| Characterization of the Ni/n-Zintl and Mg₂Ni/n-Zintl thermoelectric contact structures. (a) The fitting curve of ρ_c of Mg₂Ni junctions and the change of ρ_c of Ni junctions aging at 673 K. (b) Plots of σ_s versus the root mean square of the aging time at 673 K.

Comments (2-4):

From Supplementary Fig. 9, it can be seen that a reaction layer forms between the n-type Mg_{3.15}Co_{0.05}SbBi_{0.99}Se_{0.01} and Mg₂Ni, and the thickness of this reaction layer changes over time. What is the chemical composition of this reaction layer? And how does it affect the contact resistivity?

Response: A thinner interlayer (IL) composed of Mg-Sb(Bi)-Ni exists at the interface of the prepared junction. And its thickness basically did not change after 100 hours of aging. The chemical composition of the interlayer is detected by EDS analysis, as shown in Table R2. Without considering other factors, the contact resistivity (ρ_c) is composed of interface electrical loss and intermediate layer loss². If the thickness is predicted using diffusion theory, the fit between the contact resistivity and the thickness of the interlayer is linear, as shown in Fig. R12(b). The contact resistivity of interface electrical loss is a fixed value of 1.38 $\mu\Omega$ cm², which represents the influence of the heterogeneous interface contact barrier. The contact resistivity of intermediate layer loss is linear with the thickness of the intermediate layer, which increases linearly during 0 to 100 hours of aging and has no significant change in the subsequent 700-hour aging (Fig. R12(c-d)).

Table R2| Chemical compositions of interlayer determined by EDS.

Element composition	Mg (at.%)	Sb (at.%)	Bi (at.%)	Ni (at.%)
1	64.16	12.48	13.37	9.99
2	64.76	13.13	13.53	8.58
3	64.19	12.16	13.83	9.83
4	64.33	12.48	13.17	10.02
5	65.59	13.02	12.59	8.80

6	64.79	13.62	12.94	8.66
AVG	64.64	13.62	12.94	9.31

Fig. R12| Schematic diagram of interface resistance and result analysis. (a) Schematic diagram of interface resistance². (b) The fitting curve of the interfacial resistivity with the thickness of interlayer. (c) The change of ρ_c and (d) δ aging at 673 K.

Comments (2-5):

Certain literature related to Mg₃SbBi devices with high efficiency have not been included for comparison in this research (Energy Environ. Sci., 2021, 14, 6506–6513). Furthermore, the conversion efficiency reported in literature works such as Mg₃Sb₂/MgAgSb and Mg₃Sb₂/SKD is similar to that presented in this study within the temperature range of 300-600 K. The notably high efficiency achieved in this work is within the temperature range of 600-723 K. What is the primary reason for this?

Response: Thanks for your kind reminder. We have added more updated research results to the manuscript. More details can be found in Fig.1d and 1f (as shown in Fig. R13 below). The conversion efficiency of thermoelectric devices has a great relationship with the thermoelectric properties of materials, the temperature difference between cold and hot sides, the loss of interface structure, and the quality of device connection. In this work, the thermoelectric performance of n-type and p-type materials at 300-500 K is not much different from that of Mg₃Sb₂/SKD device reported in the

literature⁸, and the theoretical conversion efficiency of both is close in this temperature range (Fig. R14(a) and 14(b)). Therefore, under the good interface structure, the measured conversion efficiency also accords with the trend of theoretical efficiency (Fig. R14(c)). However, with the increase in temperature, the gap between the properties of materials increases. After more than 500 K, there is a large gap between the measured efficiency of the two, which is consistent with the theoretical prediction (Fig. R14(d)). The thermoelectric property of the p-type MgAgSb material in Liu's work²¹ is superior to the p-type Zintl material in this work, and its theoretical efficiency has great advantages. However, due to the unsatisfactory device connection process (the internal resistance of the device increases due to liquid metal connection), the measured conversion efficiency has decreased. The p-type MgAgSb materials cannot be used at temperatures higher than 550 K, limiting the TE modules' working temperature.

Fig. R13| Maximum conversion efficiency and raw material cost compared with the results of the same type of study. (a) Maximum conversion efficiency (η_{\max}) as a function of hot-side temperature for the two-pair module. Literature data from other single-stage modules¹⁻⁸. (b) Comparisons of the price and η_{\max} of different TE modules^{1-8,22-24}.

Fig. R14| Comparison of materials' thermoelectric properties and modules' efficiency of this work and literature works^{8,21}. (a) Comparison of thermoelectric properties of (a) n-type materials and (b) p-type materials. Comparison of (c) simulated theoretical conversion efficiency and (d) measured conversion efficiency of this work and literature works.

References:

1. Jiang, B. et al. High figure-of-merit and power generation in high-entropy GeTe-based thermoelectrics. *Science* **377**, 208-213, (2022).
2. Chu, J. et al. Electrode interface optimization advances conversion efficiency and stability of thermoelectric devices. *Nat. Commun.* **11**, 2723, (2020).
3. Jia, B. et al. Realizing high thermoelectric performance in non-nanostructured n-type PbTe. *Energy Environ. Sci.* **15**, 1920-1929, (2022).
4. Ying, P. et al. A robust thermoelectric module based on MgAgSb/Mg₃(Sb,Bi)₂ with a conversion efficiency of 8.5% and a maximum cooling of 72 K. *Energy Environ. Sci.* **15**, 2557-2566, (2022).
5. Liu, R. et al. Thermal-inert and ohmic-contact interface for high performance half-Heusler based thermoelectric generator. *Nat. Commun.* **13**, 7738, (2022).
6. Jiang, M. et al. High-efficiency and reliable same-parent thermoelectric modules using Mg₃Sb₂-based compounds. *Natl. Sci. Rev.* **10**, nwad095, (2023).
7. Bu, Z. et al. An over 10% module efficiency obtained using non-Bi₂Te₃ thermoelectric materials for recovering heat of <600 K. *Energy Environ. Sci.* **14**,

- 6506-6513, (2021).
8. Fu, Y. et al. Mg₃(Bi,Sb)₂-based thermoelectric modules for efficient and reliable waste-heat utilization up to 750 K. *Energy Environ. Sci.* **15**, 3265-3274, (2022).
 9. Shang, H. et al. N-type Mg₃Sb_{2-x}Bi_x with improved thermal stability for thermoelectric power generation. *Acta Mater.* **201**, 572-579, (2020).
 10. Paliwal, M. & Jung, I.-H. Thermodynamic modeling of the Mg-Bi and Mg-Sb binary systems and short-range-ordering behavior of the liquid solutions. *Calphad* **33**, 744-754, (2009).
 11. Dilner, D., Kjellqvist, L. & Selleby, M. Thermodynamic assessment of the Fe-Ca-S, Fe-Mg-O and Fe-Mg-S systems. *J. Phase Equilib. Diffus.* **37**, 277-292, (2016).
 12. Pei, B., Björkman, B., Sundman, B. & Jansson, B. A thermodynamic assessment of the iron-antimony system. *Calphad* **19**, 1-15, (1995).
 13. Gierlotka, W., Łapsa, J. & Jendrzeczyk-Handzlik, D. Thermodynamic description of the Pb-Te system using ionic liquid model. *J. Alloys Compd.* **479**, 152-156, (2009).
 14. Vaajamo, I. & Taskinen, P. A thermodynamic assessment of the iron-lead binary system. *Thermochim. Acta* **524**, 56-61, (2011).
 15. Arvhult, C. M., Guéneau, C., Gossé, S. & Selleby, M. Thermodynamic assessment of the Fe-Te system. Part II: Thermodynamic modeling. *J. Alloys Compd.* **767**, 883-893, (2018).
 16. Niu, C. et al. A thermodynamic assessment of the Bi-Mg-Sn ternary system. *Calphad* **39**, 37-46, (2012).
 17. Jacobs, M. & Spencer, P. A critical thermodynamic evaluation of the system Mg-Ni. *Calphad* **22**, 513-525, (1998).
 18. Li, Q. et al. Experimental investigation and thermodynamic modeling of the phase equilibria at the Mg-Ni side in the La-Mg-Ni ternary system. *J. Alloys Compd.* **509**, 2478-2486, (2011).
 19. Miettinen, J. Thermodynamic description of Cu-Mg-Ni and Cu-Mg-Zn systems. *Calphad* **32**, 389-398, (2008).
 20. Vassilev, G., Gandova, V. & Docheva, P. Comments and reconciliation of the Ni-Bi-system thermodynamic reassessments. *Cryst. Res. Technol.* **44**, 25-30, (2008).
 21. Liu, Z. et al. Demonstration of ultrahigh thermoelectric efficiency of ~7.3% in Mg₃Sb₂/MgAgSb module for low-temperature energy harvesting. *Joule* **5**, 1196-1208, (2021).
 22. Jiang, M. et al. High-efficiency and reliable same-parent thermoelectric modules using Mg₃Sb₂-based compounds. *Natl. Sci. Rev.*, nwad095, (2023).
 23. Zhu, B. et al. Realizing record high performance in n-type Bi₂Te₃-based thermoelectric materials. *Energy Environ. Sci.* **13**, 2106-2114, (2020).
 24. Yu, J. et al. Half-Heusler thermoelectric module with high conversion efficiency and high power density. *Adv. Energy Mater.* **10**, 2000888, (2020).

REVIEWERS' COMMENTS

Reviewer #1 (Remarks to the Author):

The authors have addressed all questions, I suggest to accept it.

Reviewer #2 (Remarks to the Author):

The authors have devoted much effort to enhance the quality of this study, resulting in a significant improvement. The revised version effectively emphasized the novelty and reliability of this research, including the CALPHAD design method, advanced device preparation technology, high thermoelectric performance and conversion efficiency. I believe that the potential applicability of this strategy to other thermoelectric materials will capture the interest of readers. I am pleased to accept this paper.

Response to reviewers

Reviewer #1:

The authors have addressed all questions, **I suggest to accept it.**

Response: We thank the reviewer very much. We are delighted that the improved manuscript addresses the reviewer's concerns.

Reviewer #2:

The authors have devoted much effort to enhance the quality of this study, resulting in a significant improvement. **The revised version effectively emphasized the novelty and reliability of this research, including the CALPHAD design method, advanced device preparation technology, high thermoelectric performance and conversion efficiency.** I believe that the potential applicability of this strategy to other thermoelectric materials will capture the interest of readers. **I am pleased to accept this paper.**

Response: We thank the reviewer very much for the valuable suggestions that have greatly helped us to improve the manuscript.